# Influence of Marine Yeast *Debaryomyces hansenii* on Antifungal and Physicochemical Properties of Chitosan-Based Films

**DOI:** 10.3390/jof8040369

**Published:** 2022-04-04

**Authors:** César A. García-Bramasco, Francisco J. Blancas-Benitez, Beatriz Montaño-Leyva, Laura M. Medrano-Castellón, Porfirio Gutierrez-Martinez, Ramsés R. González-Estrada

**Affiliations:** 1Tecnológico Nacional de México/Instituto Tecnológico de Tepic, Avenida Tecnológico 2595, Col. Lagos del Country, Tepic 63175, Nayarit, Mexico; ceargarciabr@ittepic.edu.mx (C.A.G.-B.); lamamedranoca@ittepic.edu.mx (L.M.M.-C.); pgutierrez@ittepic.edu.mx (P.G.-M.); 2Departamento de Investigación y Posgrado en Alimentos, Universidad de Sonora, Rosales y Blvd. Luis Encinas, C.P., Hermosillo 83000, Sonora, Mexico; beatriz.montano@unison.mx

**Keywords:** biocontrol potential, cell viability, chitosan films, *Penicillium italicum*

## Abstract

Chitosan-based film with and without antagonistic yeast was prepared and its effect against *Penicillium italicum* was evaluated. The biocompatibility of yeast cells in the developed films was assessed in terms of population dynamics. Furthermore, the impact on physicochemical properties of the prepared films with and without yeast cells incorporated were evaluated in terms of thickness, mechanical properties, color and opacity. Chitosan films with the antagonistic yeast entrapped exhibited strong antifungal activity by inhibiting the mycelial development (55%), germination (45%) and reducing the sporulation process (87%). Chitosan matrix at 0.5% and 1.0% was maintained over 9 days of cell viability. However, at 1.5% of chitosan the population dynamics was strongly affected. The addition of yeast cells only impacted color values such as a*, b*, chroma and hue angle when 1.0% of chitosan concentration was used. Conversely, luminosity was not affected in the presence of yeast cells as well as the opacity. Besides, the addition of antagonistic yeast improved the mechanical resistance of the films. The addition of *D. hansenii* in chitosan films improve their efficacy for controlling *P. italicum,* and besides showed desirable characteristics for future use as packaging for citrus products.

## 1. Introduction

Nowadays, the preservation of food is crucial to respond to the high demand of healthy foods as well as consumers’ preference for produce with longer shelf-life [1]. An alternative is the formulation and use of edible packaging films not only for food protection but also as a strategy for using eco-friendly materials in the food industry [2]. In this sense, chitosan is a versatile biopolymer derived from natural resources, biocompatible, with good film-forming ability and antimicrobial activity [3,4]. It is recognized that edible films can be used as carriers of several compounds such as plant extracts [5], colors and flavors [6], essential oils [7], probiotics [8] and microbial antagonists [9,10]. Several studies reported the carrier capacity of chitosan of several compounds such as gallic acid [11], nisin [12], sulfur nanoparticles [13] and essential oils [14], among others. However, to our knowledge, there is no report about the incorporation of marine yeasts *Debaryomyces hansenii* in chitosan-based films. 

On the other hand, antagonists have been used successfully for controlling important pathogens such as *Colletotrichum gloeosporioides* [15,16], *Fusarium proliferatum* [17], *Botrytis cinerea* [18], *Penicillium digitatum* [19] and *Penicillium italicum* [20]. Marine yeast *Debaryomyces hansenii* can control phytopathogens throughout several mechanisms of action such as the synthesis of hydrolytic enzymes (chitinase, glucanase, protease), competition for space and nutrients, production of *killer* toxins and biofilms onto the fruit surface, thus the use of antagonistic *D. hansenii* to manage fungal diseases is a promising alternative to chemical management [21,22,23].

It is reported that edible films can improve the biocontrol activity of microbial antagonists [10,24]. In this context, in a recent study, the antagonistic yeast *Meyerozyma caribbica* was added into sodium alginate-based films; the results showed the inhibition of *Colletotrichum gloeosporioides* not only in the mycelial growth (100%) but also in the germination process (99%) [25]. In another study, Aloui et al. [26] reported that sodium alginate-based films with the antagonistic yeast *Wickerhamomyces anomalus* entrapped can control *Penicillium digitatum* in vitro, by reducing the mycelial development. The objective of this study was to evaluate the impact of the addition of the antagonistic yeast *D. hansenii* in chitosan-based films in terms of cell viability (population dynamics), antifungal efficacy against *P. italicum* and physicochemical properties of films.

## 2. Materials and Methods

### 2.1. Raw Materials

Commercial chitosan (Molecular weight = 3.01 × 10^4^ Da, cp = 200, 90% of deacetylation, Golden-Shell Pharmaceutical Co., Ltd., Zhejiang, China) was used. Glycerol from Sigma-Aldrich (St. Louis, MO, USA) was used as a plasticizer. Glacial acetic acid and Tween 80 were purchased from Jalmek (San Nicolás de los Garza, Mexico).

### 2.2. Microorganisms

The antagonistic yeast and the pathogen *Penicillium italicum* (isolated from Persian lime) were provided by CIBNOR S.C. The yeast was maintained on yeast extract peptone dextrose (YPD) medium [27]. The fungus was maintained on PDA (Potato dextrose agar) plates (Sigma-Aldrich, St. Louis, MO, USA).

### 2.3. Film Preparation

Chitosan-based films were prepared by a casting method following the protocol proposed by Homez-Jara et al. [28] with some modifications. Chitosan-based film forming solutions were prepared by dissolving 0.5%, 1.0% or 1.5% (*w*/*v*) in 1% of glacial acetic acid, glycerol (0.3% *v*/*v*) and Tween 80 (0.1% *v*/*v*) with constant agitation using magnetic stir at room temperature for 180 min. The chitosan solutions were vacuum filtered and allowed to stand overnight. The film forming solutions (without yeast) were sterilized at 121 °C during 15 min. For the films incorporated with the antagonistic yeast, firstly a solution containing the yeast was prepared: briefly, in a biosafety hood (Novatech BBS-DDC) Petri dishes containing yeasts were scrapped with a sterile inoculation loop, then the yeasts were incorporated into a sterile saline solution (0.85%) and adjusted to 1 × 10^7^ cell/mL. Then, the solution was centrifuged at 6000 rpm for 20 min, the resulting pellet was aseptically removed and incorporated to the sterile chitosan-based film forming solution in sterile conditions. Finally, the film forming solutions with and without yeast were poured into Petri dishes (60 × 15 mm) and they were left to dry for 40 h in an oven at 40 °C (22% of relative humidity).

### 2.4. Antifungal Test

The mycelial growth inhibition was performed using the protocol proposed by González-Estrada et al. [27] with some modifications. Firstly, the fungus was grown on PDA plates for 7 days (28 °C), then with a sterile inoculation loop; a portion of the fungus mycelia was taken and inoculated in the center of PDA Petri dishes. Thereafter, the surface was covered with chitosan-based films containing or not the antagonistic yeast. Finally, plates were incubated at 28 °C for 6 days, the colony diameter was registered daily using a digital Vernier (Truper S.A. de C.V.™, Villahermosa, Mexico). Control plates consisted of PDA plates without films. The results were registered as percentage of mycelial growth inhibition according to the following formula [27]:Inhibition (%) = [(dc − dt)/dc] × 100.
dc (cm) is the mean of colony diameter for the control and dt (cm) is the mean of the colony diameter of treatments. Once the mycelial inhibition test was finished, the same Petri dishes were used for the sporulation test. Briefly, the film was removed aseptically, then sterile distilled water (10 mL) was added to the fungal lawn and the film that was in contact with the colony and was rubbed with an inoculation loop in order to release the spores. The mycelia present in the suspension was retained using a sterile cheesecloth. The spore concentration (number of spores/mL) was calculated using a hemocytometer (Hausser Scientific^®^, Horsham, PA, USA), the percentage of spore reduction was calculated considering the spore’s concentration of the control plate. A germination test was performed using the protocol proposed by González-Estrada [27] with some modifications. Briefly, 20 µL of the spore suspension (1 × 10^6^ spores/mL) was added into PDA plugs (6 mm), the inoculum was left to dry for 15 min in a biosafety hood, then the plugs were covered with the films containing the antagonistic yeast or not. Control samples consisted of PDA plugs without films. After 12 h of sample incubation the films were removed aseptically in order to quantify the germinated spores using a microscope (Motic, BA300, San Antonio, TX, USA) by counting 200 spores per sample as a reference.

### 2.5. Viability of D. hansenii in Films

For the evaluation of yeast’s viability the protocol proposed by González-Estrada et al., (2017) was applied with some modifications: films were aseptically cut and placed into a sterile saline solution (0.85%), stirred (350 rpm) at room temperature for 48 h until the film was dissolved in order to release the cells from the film. Serial dilutions were made and 100 µL of sample was inoculated on YPD and incubated for 24 h at 28 °C, and finally the number of viable yeasts were counted. The results were expressed as Log CFU/film.

### 2.6. Film Characterization: Physicochemical Properties

#### 2.6.1. Thickness

The thicknesses of each film were measured using a digital micrometer (Fowler Electronic Micrometer, Newton, MA, USA) at ten randomly selected points on each sample [29].

#### 2.6.2. Mechanical Properties

The tensile test evaluated in the films were stress at break (MPa), strain at break (%) and Young’s Modulus (MPa), following the method ASTM D-88-02 [30]. Stress at break is the maximum stress supported by a material and is estimated by dividing the maximum load by the cross-sectional area of the initial sample. Strain at break is the percentage change in the length of the sample from the original length (L) between grips (20 mm). Young’s modulus is calculated from the initial linear slope of the stress-strain curve. The mechanical properties were evaluated in a texture analyzer (TAXT plus, Vienna Court, Surrey, UK). Films were cut into 4.0 × 0.5 cm strips and were analyzed under a tension force with a load cell of 50 N and a crosshead speed of 1 mm/min.

#### 2.6.3. Optical Properties

##### Color Assessments

Chitosan-based films color was evaluated with a colorimeter (PCE-CSM 7). The color coordinates, L (lightness), a* (red–green) and b* (yellow–blue) were used to determine the Chroma ($C_{ab}^{*}$) and hue angle ($h_{ab}^{*}$) and color change $\left(\Delta E^{*}\right)$, induced by the presence of yeast in films, using the following equations: $$C_{ab}^{*}=\sqrt{a^{*2}+b^{*2}}$$
$$h_{ab=}^{*}\mathrm{arctg}\left(\frac{b^{*}}{a^{*}}\right)$$
$$\Delta E^{*}=\sqrt{{\left(\Delta L\right)}^{2}+{\left(\Delta a^{*}\right)}^{2}+{\left(\Delta b^{*}\right)}^{2}}.$$

##### Opacity

The opacity was calculated based on average thickness. This determination was evaluated following the protocol proposed by Souza et al. [29], using rectangular samples from each film and using a UV/VIS spectrophotometer (Thermo Scientific GENESYS 10S, Waltham, MA, USA) at 600 nm, the opacity was calculated using the following formula:$$Opacity\;({\text{mm}}^{-1})=\frac{Absorbance\;600\;\text{nm}}{film\;thickness\;\left(\text{mm}\right)}.$$

### 2.7. Statistical Analysis

A factorial design (2 × 3) was applied for the analysis of data for in vitro tests, thickness and optical properties, taking into account the presence or not of antagonistic yeast and chitosan concentration (0.5, 1.0 and 1.5%). For mycelial growth and sporulation assays five Petri dishes were used and for germination assay five PDA disks were used per replicate. For thickness and optical properties, three films were used per replicate. Uni-factorial statistical design was applied for population dynamics in films, three films were used per replicate. All the experiments were repeated twice. The Analysis of variance (ANOVA) with a statistical significance of 5% was conducted using Minitab Statistical Software^®^ (version 19.2020.1.0, Minitab, LLC, Chicago, IL, USA). The LSD test was used to determine significant differences among chitosan concentration and the presence or not of antagonistic yeast.

## 3. Results

### 3.1. Antifungal Test

The antifungal activity of elaborated films against *P. italicum* is shown in Figure 1 and Figure 2. Overall, efficacy of films was significant (*p* < 0.05) depending on the chitosan concentration and the presence or not of the antagonistic yeast. The application of chitosan-based films at 1.5% with yeast was more efficient to control mycelial development (55%) compared to films without yeasts (37%).

Germination of *P. italicum* was mostly inhibited with the application of chitosan-based films (1.5%) with antagonistic yeasts entrapped (45%) compared to films without yeasts (30%).

The sporulation process was strongly affected at all concentrations of chitosan evaluated in combination with the antagonistic yeast (Figure 2c,d). For films with the yeast, entrapped lower values of spores/mL were reported (4.5 × 10^6^), reducing the sporulation process by up to 81%; conversely, chitosan films without yeast reported 3.2 × 10^7^ of spores/mL and a sporulation reduction of 12%.

### 3.2. Population Dynamics of Yeast in Films

The effect of chitosan on the growth of antagonistic yeast was evaluated on the population dynamics of *D. hansenii* in films. Depending on chitosan concentration, the viability of yeast was decreasing over the storage time (Figure 3). On the 6th day with 0.5% of chitosan the population dynamics was maintained 100%. Conversely, with 1.0% of chitosan, 75% of the initial inoculum was preserved. On the 9th day the inoculum was maintained at 62% and 24%, for 0.5% and 1.0% of chitosan respectively. Conversely, with 1.5% of chitosan on the sixth day no viability was evidenced.

### 3.3. Film Characterization: Physicochemical Properties

The basic film properties, including thickness, mechanical properties, color and opacity, are shown in Table 1 and Table 2. Film thickness was only significantly affected (*p* < 0.05) by the chitosan concentration, as previously reported [28], but not for the addition of yeast cells. Film luminosity decreased as the chitosan concentration increased, as previously reported [31]. Conversely, the incorporation of yeasts had no effect on the film’s lightness. Parameter a* had positive values, which means that a red color was noted in lesser or greater intensity depending on the chitosan concentration. Besides, only significant changes were noted in this parameter in films at 1.0% of chitosan with yeast (3.4) compared to films without yeast (2.9). The b* values significantly increased according to the chitosan concentration (*p* < 0.05), indicating that the films were yellower (Figure 4), as previously reported [28]. Besides, only changes in b* values were noted by the addition of yeasts (12.9) compared to films without yeasts (9.7) at 1.0% of chitosan concentration. Hence, chitosan films with and without yeasts became slightly reddish and yellowish, but they were still transparent (Figure 4). For chroma values, an increase according to chitosan concentration was noted, whereas the incorporation of *D. hansenii* in the polymeric matrix was only significant (*p* < 0.05) at 1.0% of chitosan concentration. The same behavior was noted for hue angle in films with yeasts (at 1.0% of chitosan), whereas at 0.5% and 1.5% of chitosan the addition of yeast did not change this parameter. Significant changes (*p* < 0.05) were noted on $\Delta E^{*}$ depending on chitosan concentration; conversely, the presence of yeast only impacted this value at low chitosan concentrations (0.5%). The opacity is an inverse measure of transparency; thus, the transparency decreases as its opacity increases. The opacity values were observed range from 2.5 to 4.82 mm^−1^ in films without yeasts and 3.3 to 7.4 mm^−1^ in films with yeasts.

The effects of incorporating yeast on the tensile properties of chitosan films are presented in Table 1. According to statistical analysis results, both the concentration and the yeast addition showed a significant effect for stress at break and Young’s modulus (*p* < 0.0001). Both parameters increased with the concentration of chitosan in the films. It is observed that when the yeasts were added to the films, the stress at break increased by nearly double; this effect was more evident at a higher concentration of chitosan (1.5% chitosan, 10.92 ± 1.68 MPa). Similarly, Young’s modulus increased significantly when yeasts were incorporated into the chitosan matrix. For the maximum chitosan concentration (1.5%), Young’s modulus increased by around 250% (6.51 ± 0.76 MPa) compared to the film without yeast (2.54 ± 0.37 MPa).

Strain at break also increased significantly with higher chitosan concentration. However, the addition of yeast into the films did not show a significant effect (*p* = 0.0686). In Table 1 it is observed that in chitosan-yeast films (1.5%) the strain at break was 3.98%, which corresponds to a decrease of 29% compared to the chitosan films (without yeast, 5.14 ± 0.55%).

## 4. Discussion

Chitosan has the ability to induce the permeabilization of cell membranes leading to the loss of cell content, besides this interaction causes biochemical alterations such as protein biosynthesis, carbohydrate metabolism and energy production in the fungus affecting the mycelial development [32,33]. On the other hand, *D. hansenii* has the ability to produce lytic enzymes (protease, glucanase and chitinase) which are related to their antifungal activity; these enzymes can interact with the cell wall of *P. italicum* causing their degradation [23,34]. The results showed a better efficacy by the combination of chitosan with the antagonistic yeast (Figure 1), previous studies have reported a synergistic effect of chitosan against pathogens when the biopolymer is combined with antagonistic yeasts, due to a better distribution of the antagonistic cells that can be in contact with the pathogen allowing the production of lytic enzymes, as previously reported [27,35,36,37,38].

Chitosan inhibition could be related to the electrostatic forces between the amino groups (-NH_2_) of chitosan with the negative charges on the cell membrane of the spore, causing changes in its permeability and alterations at the intracellular level, affecting the germination process [3]. Nevertheless, the efficacy was better for films with the *D. hansenii* entrapped, in this sense two mechanisms of action such as hydrolytic enzymes and nutrient competition are related to the affectation of spore development of *P. italicum* avoiding the tube elongation, as previously reported [27,39]. Our results are in agreement with those reported by Wang et al. [36]. In their investigation, the combination of carboxymethyl chitosan with *Cryptococcus laurentii* was more efficient for inhibiting not only the germination process of *P. italicum* but also the germ tube length than a single application of chitosan.

The results of the sporulation can be related to the damage in the mycelia, even when with chitosan films at 0.5% and 1.0% and with the antagonistic yeast entrapped, lower values of mycelial development were reported (Figure 2a), the results suggest a significant damage to fungus structure that affects spore production. *P. italicum* is considered one of the most devastating pathogens for citrus fruits, causing blue mold decay [40]. Germination and sporulation as well as mycelial growth play a key role in the infection cycle of *P. italicum* in citrus, thus any treatment with antifungal activity is crucial to affect the establishment of the fungus in susceptible fruits [41].

In the population dynamics our results are in agreement with previous reports, the same effect has been evidenced for *Cryptococcus laurentii* exposed to carboxymethyl chitosan [36] and chitosan [42]. However, in both studies the exposition with chitosan was temporal (24 h), thus our results are better due to the maintenance of viability for 5 days more and even at high concentrations of chitosan.

The study of mechanical properties in coatings and food packaging is extremely important since the packaging must resist mechanical stress to protect the food and be able to maintain its integrity during handling. Chitosan composition and intermolecular forces play an important role in the mechanical properties of chitosan-based films [43]. It is also important to consider the compatibility of the chitosan with the other components in the film. The increase in stress at break is related to good interaction between polymers, improving the mechanical resistance of the films. Likewise, increasing elongation results in better ductility. Koc et al. [44] studied the mechanical properties in chitosan-fungal extract films (*Trichholoma terreum*). The elasticity of the chitosan films was increased with the addition of the fungal extract. On the contrary, tensile strength and Young’s modulus decreased. This behavior was attributed to the interaction between the chitosan polymer chains with the fungal extract molecules.

Similarly, González-Estrada et al. [9] added *D. hansenni* yeast to covalently cross-linked arabinoxylans films. The evaluation of the mechanical properties of the films showed that tensile strength, elongation at break and Young’s modulus values decreased when *D. hansenii* was added to the film. They reported that changes in the mechanical properties were due to defects in the arabinoxylans films, which contribute to an early rupture of the films during tensile tests.

β-glucans are the main structural components of the cell wall in yeasts. The *D. hansenni* cell wall was characterized by Medina-Córdova et al. [45] using NMR. They found structures containing (1-6)-branched (1-3)-β-D-glucan. The interaction between chitosan and β-glucans in films has been previously reported. Koc et al. [44] studied the interactions between a fungal extract and polyphenols in a chitosan matrix. FTIR spectra indicated that the OH absorption peak was broadened and shifted towards lower frequencies due to the formation of hydrogen bonds between the chitosan film matrix, β-glucan and polyphenolic compounds. In our study, stress at break increased with the addition of yeast, which could indicate the presence of interactions and good compatibility between molecules (e.g., chitosan molecules and yeast β-glucans). Thus, enhancing mechanical resistance of the films. The decrease in strain at break showed by films with 1.5% chitosan + yeast might be due to the higher number of interactions between the yeast and the functional groups in chitosan when the latter polymer is at a higher concentration. Thus, the free volume and molecular mobility of the polymer chains are decreased. This behavior has been previously observed in chitosan films incorporated with essential oils [46,47] and polyphenols [43]. In this work, the concentration of chitosan and the addition of yeasts improved the mechanical properties of the films. Mechanical resistance was improved while maintaining ductility, therefore these materials could be used as edible films to control the growth of pathogens and improve the safety and quality of food.

In color parameters, our results are in agreement with González-Estrada et al. [9], whose study reported changes in color values such as a* and b* in arabinoxylan films with *D. hansenii* entrapped. The opacity and $\Delta E^{*}$ values showed that the addition of yeasts only impacted films with a low concentration of chitosan (0.5%), conversely at 1.0% and 1.5% of chitosan no changes were observed. The opacity values of this study agreed with those reported by Homez-Jara et al. [28], ranging from 2.4 to 5.1 mm^−1^. Color and transparency of films are crucial factors of packaging materials in which the visual characteristic can impact consumer acceptance [48]. Future research is needed considering the use of chitosan combined with the antagonistic yeast as a coating on citrus.

## 5. Conclusions

In conclusion, chitosan combined with *D. hansenii* proved to be more successful at inhibiting the growth of fungi in vitro than the application of chitosan alone. At 0.5% and 1.0% of chitosan concentration, the viability of yeast was maintained for 9 days. The incorporation of antagonistic yeast improved the mechanical resistance of the films. Chitosan in combination with yeast not only achieves the viability of the antagonist, but also shows important effects on parameters involved in the development and proliferation of the fungus. In addition, the films have desirable characteristics for packaging use for citrus in the future. Nevertheless, the combined use of chitosan at 0.5% and 1.0% with the antagonistic yeast needs to be tested in in vivo trials on citrus fruit, in order to determine if the resulting population dynamics is sufficient to prevent fungal infection by *Penicillium italicum*.

## Figures and Tables

**Figure 1 jof-08-00369-f001:**
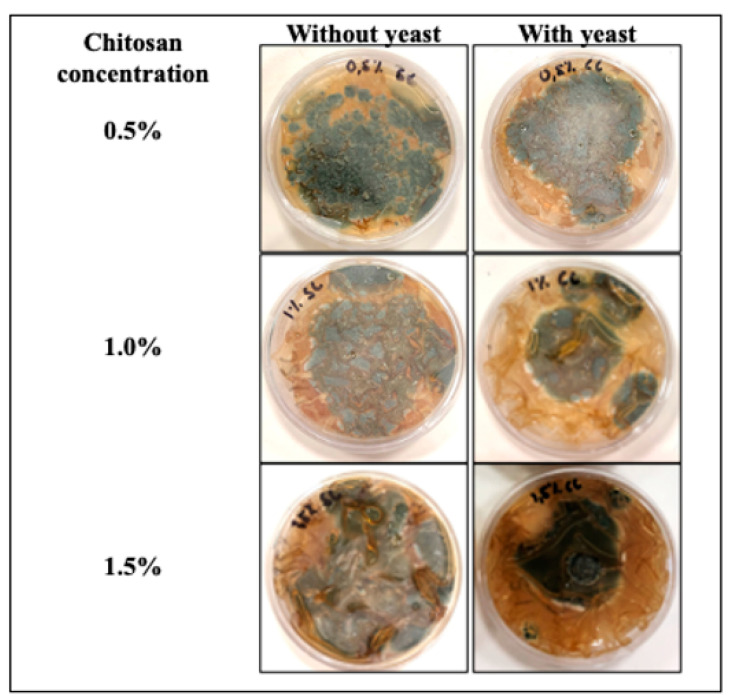
Mycelial development of *Penicillium italicum* exposed to chitosan-based films for 6 days.

**Figure 2 jof-08-00369-f002:**
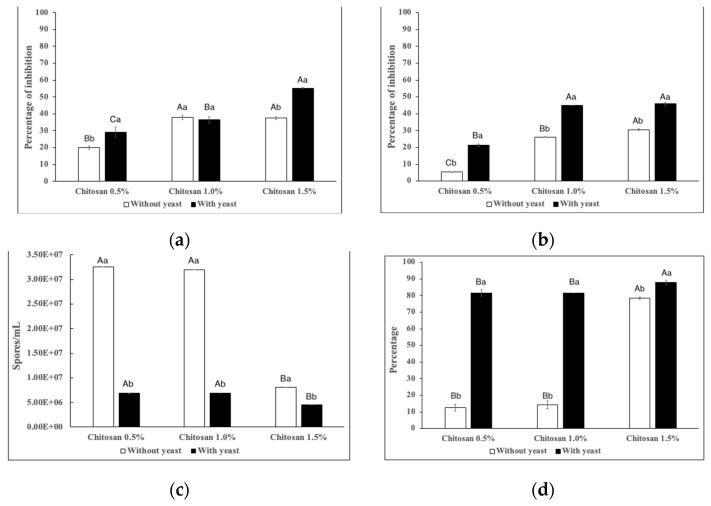
Effect of chitosan-based films with or without antagonistic yeasts against *Penicillium italicum* development, (**a**) mycelial growth, (**b**) germination, (**c**) sporulation and (**d**) sporulation reduction. According to the LSD test, different letters indicate significant differences (*p* < 0.05) among chitosan concentration (upper-case) and presence or not of antagonistic yeast (lower-case). Values are expressed as means ± standard deviation.

**Figure 3 jof-08-00369-f003:**
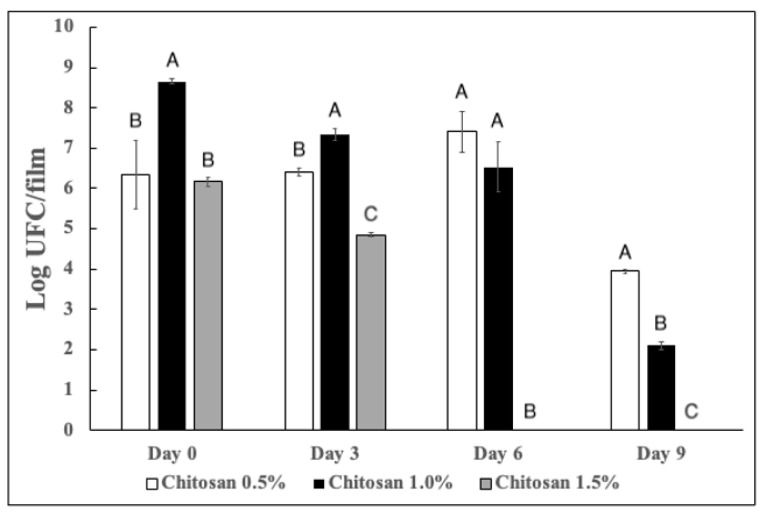
Population dynamics of *D. hansenii* entrapped in chitosan films. According to the LSD test, different letters indicate significant differences (*p* < 0.05). Values are expressed as means ± standard deviation.

**Figure 4 jof-08-00369-f004:**
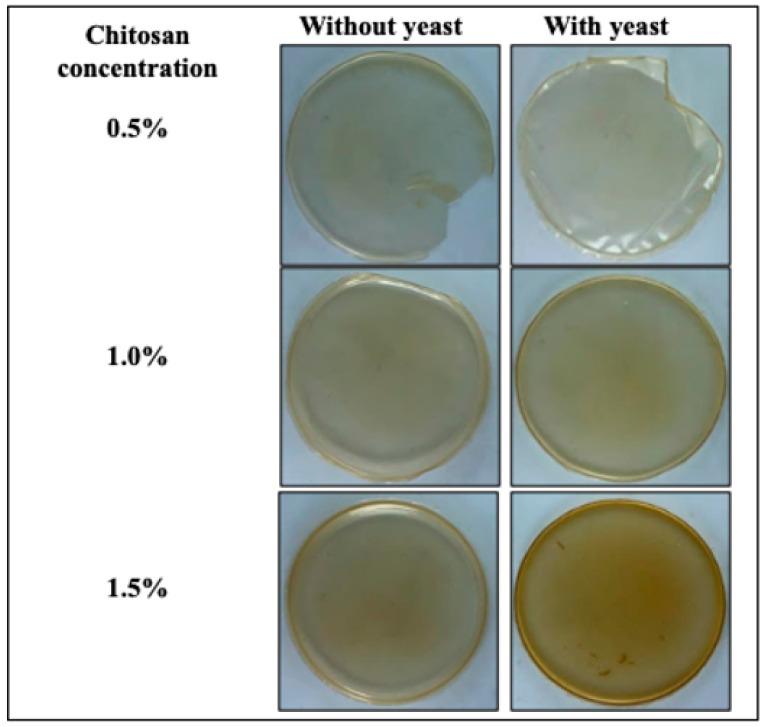
Chitosan-based films with or without marine yeast.

**Table 1 jof-08-00369-t001:** Thickness and tensile properties of chitosan films with or without yeasts.

	Thickness (µm)		Stress at Break (MPa)		Strain at Break (%)		Young Modulus (MPa)	
Chitosan concentration	Without yeast	With yeast	Without yeast	With yeast	Without yeast	With yeast	Without yeast	With yeast
0.5%	24.7 ± 0.42 ^Ca^	52.5 ± 10.3 ^ABa^	1.88 ± 0.13 ^Aa^	3.55 ± 0.31 ^Ab^	1.25 ± 0.14 ^Aa^	2.09 ± 0.44 ^Aa^	1.24 ± 0.27 ^Aa^	1.85 ± 0.18 ^Ab^
1.0%	35.9 ± 5.52 ^Ba^	40.9 ± 0.42 ^Ba^	2.84 ± 0.51 ^ABa^	9.71 ± 0.40 ^Bb^	1.51 ± 0.23 ^ABa^	3.05 ± 0.41 ^Bb^	1.86 ± 0.22 ^Ba^	4.79 ± 0.19 ^Bb^
1.5%	66.1 ± 0.14 ^Aa^	70.2 ± 3.68 ^Aa^	4.83 ± 0.81 ^Ca^	10.92 ± 1.68 ^Cb^	5.14 ± 0.55 ^Cb^	3.98 ± 0.90 ^Bb^	2.54 ± 0.37 ^Ca^	6.51 ± 0.76 ^Cb^

Different letters indicate significant differences (*p* < 0.05), among the evaluated chitosan concentration (upper-case) and presence or not of antagonistic yeast (lower-case). Values are expressed as means ± standard deviation (*n* = 3).

**Table 2 jof-08-00369-t002:** Optical properties of chitosan films with or without yeasts.

	L*		a*		b*		$C_{ab}^{*}$		$h_{ab}^{*}$		$\Delta E^{*}$		Opacity (Abs_600_ mm^−1^)	
Chitosan concentration	Without yeast	With yeast	Without yeast	With yeast	Without yeast	With yeast	Without yeast	With yeast	Without yeast	With yeast	Without yeast	With yeast	Without yeast	With yeast
0.5%	78.5 ± 0.42 ^Ab^	80.5 ± 1.09 ^Aa^	1.82 ± 0.07 ^Ca^	1.84 ± 0.20 ^Ca^	7.16 ± 0.39 ^Ca^	7.04 ± 0.97 ^Ca^	7.39 ± 0.39 ^Ca^	7.28 ± 0.99 ^Ca^	75.7 ± 0.55 ^Aa^	75.2 ± 0.65 ^Aa^	17.5 ± 0.47 ^Ca^	15.6 ± 1.27 ^Cb^	4.82 ± 0.03 ^Bb^	7.45 ± 0.22 ^Aa^
1.0%	75.5 ± 1.13 ^Ba^	75.8 ± 0.37 ^Ba^	2.97 ± 0.28 ^Bb^	3.41 ± 0.34 ^Ba^	9.78 ± 0.93 ^Bb^	12.9 ± 1.63 ^Ba^	10.2 ± 0.96 ^Bb^	13.3 ± 1.66 ^Ba^	73.0 ± 0.87 ^Bb^	75.1 ± 0.40 ^Aa^	21.4 ± 0.90 ^Ba^	22.8 ± 1.11 ^Ba^	5.53 ± 0.29 ^Aa^	5.32 ± 0.66 ^Ba^
1.5%	72.9 ± 1.09 ^Ca^	72.9 ± 1.05 ^Ca^	5.69 ± 0.46 ^Aa^	5.99 ± 0.58 ^Aa^	19.2 ± 0.90 ^Aa^	18.9 ± 1.30 ^Aa^	20.0 ± 0.99 ^Aa^	19.9 ± 1.39 ^Aa^	73.5 ± 0.65 ^Ba^	72.5 ± 0.83 ^Bb^	29.2 ± 0.70 ^Aa^	29.12 ± 1.59 ^Aa^	2.55 ± 0.20 ^Ca^	3.35 ± 0.79 ^Ca^

Different letters indicate significant differences (*p* < 0.05), among the evaluated chitosan concentration (upper-case) and presence or not of antagonistic yeast (lower-case). Values are expressed as means ± standard deviation (*n* = 3). * Part of the nomenclature for color values.

## Data Availability

The datasets generated during the study are availabe from the corresponding authors upon reasonable request.

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
