# Peer review of "Influence of Marine Yeast Debaryomyces hansenii on Antifungal and Physicochemical Properties of Chitosan-Based Films"

_jof, 2022, doi:10.3390/jof8040369_

Round 1

Reviewer 1 Report

- There are some grammatical errors and misspellings (such as log CFU in lines 172, 173, and y-axis of fig. 3), please revise.

  •  More information about marine yeasts should be added to the Introduction to explain why the author used this yeast.
  • Line 69, unit of yeast counting should be cell/mL, shouldn't it?
  • Line 76-78, "Firstly, the fungus was grown on PDA 76
    plates during 7 days (28 °C), then using a sterile inoculation loop a small portion of the fungus was taken and inoculated in the center of PDA Petri dishes." Please rewrite.
  • Line 100, the citation should be placed after the name of the protocol owner in line 99?
  • For mechanical testing, which standard method or protocol did the author use as a reference?
  • Line 156-158, please rewrite the sentence "Significantly (P<0.05) lower values of spores/mL were reported (4.5x106) for films with the yeast entrapped, reducing the sporulation up to 81%, conversely, chitosan films without yeast reported 3.2x107 of spores/mL and a sporulation reduction only of 12%.".
  • Images of chitosan films with and without yeast should be shown.
  • Figure 2 a-d, unit and title of y-axis must be added.
  • Line 172, slight 'increase of yeast population' was evidenced?
  • Line 226, 'membrane cells' should be 'cell membranes'?
  • As the author discussed in Line 271-272, 2750276 about the good compatibility between yeast and chitosan, and molecular mobility which related to the mechanical property of the films. How can the author prove that? Please explain more, or add more results to support and discuss.
  • In conclusion, which one is the suggested optimal formulation of film for future application?

Author Response

Response to Reviewer 1 Comments

Point 1: More information about marine yeasts should be added to the Introduction to explain why the author used this yeast.

Response 1: The following information was added : “Marine yeast Debaryomyces hansenii can control phytopathogens throughout several mechanisms of action such as synthesis of hydrolytic enzymes (chitinase, glucanase, protease), competition for space and nutrients, production of killer toxins and biofilms onto the fruit surface, thus the use of antagonistic D. hansenii to manage fungal diseases is a promising alternative to chemical management [17–19].” As well as the following references:

  1. Hernandez-Montiel, L.G.; Droby, S.; Preciado-Rangel, P.; Rivas-García, T.; González-Estrada, R.R.; Gutiérrez-Martínez, P.; Ávila-Quezada, G.D. A Sustainable Alternative for Postharvest Disease Management and Phytopathogens Biocontrol in Fruit: Antagonistic Yeasts. Plants 2021, 10, 2641.
  2. Droby, S.; Gonzalez-Estrada, R.R.; Avila-Quezada, G.; Durán, P.; Manzo-Sánchez, G.; Hernandez-Montiel, L.G. Microbial Antagonists from Different Environments Used in the Biocontrol of Plant Pathogens. In Microbial Biocontrol: Food Security and Post Harvest Management; Springer, 2022; pp. 227–244.
  3. Medina-Córdova, N.; Rosales-Mendoza, S.; Hernández-Montiel, L.G.; Angulo, C. The Potential Use of Debaryomyces Hansenii for the Biological Control of Pathogenic Fungi in Food. Biological Control 2018, 121, 216–222.

Point 2: Line 69, unit of yeast counting should be cell/mL, shouldn't it?

Response 2: The authors changed the paragragph, as suggested by the reviewer.

Point 3: Line 76-78, "Firstly, the fungus was grown on PDA 76

plates during 7 days (28 °C), then using a sterile inoculation loop a small portion of the fungus was taken and inoculated in the center of PDA Petri dishes." Please rewrite.

Response 3: The authors changed the paragragph, as suggested by the reviewer: “Firstly, the fungus was grown on PDA plates during 7 days (28 °C), then with a sterile inoculation loop; a portion of the fungus mycelia was taken and inoculated in the center of PDA Petri dishes”.

Point 4: Line 100, the citation should be placed after the name of the protocol owner in line 99?

Response 4: Changes were applied as sussgested by the reviewer.

Point 5: For mechanical testing, which standard method or protocol did the author use as a reference?

Response 5: The standard method was added as suggested by the reviewer: “The tensile test evaluated in the films were stress at break (MPa), strain at break (%) and Young’s Modulus (MPa) following the method ASTM D-88-02 (ASTM, 2001)”.

Reference: ASTM (2001) Standard test method for tensile properties of thin plastic sheeting. In: Standard D882-02, Annual book of ASTM, pp 162–170.

Point 6: Line 156-158, please rewrite the sentence "Significantly (P<0.05) lower values of spores/mL were reported (4.5x106) for films with the yeast entrapped, reducing the sporulation up to 81%, conversely, chitosan films without yeast reported 3.2x107 of spores/mL and a sporulation reduction only of 12%.".

Response 6: Changes were applied as sussgested by the reviewer. “For films with the yeast entrapped lower values of spores/mL were reported (4.5x106), reducing the sporulation process up to 81%, conversely chitosan films without yeast reported 3.2x107 of spores/mL and a sporulation reduction of 12%.”

Point 7: Images of chitosan films with and without yeast should be shown.

Response 7: The images of films were added as suggested by the reviewer

Point 8: Figure 2 a-d, unit and title of y-axis must be added.

Response 8: Missing information was added in the figure as suggested by the reviewer

Point 9: Line 172, slight 'increase of yeast population' was evidenced?

Response 9: According to the results no increased was evidenced, thus the text was changed to: “Depending of chitosan concentration, the viability of yeast was decreasing over the storage time (Fig. 3). At 6th day with 0.5% of chitosan the population dynamics was maintained 100%.”

Point 10: Line 226, 'membrane cells' should be 'cell membranes'?

Response 10: The correction was made as suggrested by the reviewers

Point 11: As the author discussed in Line 271-272, 2750276 about the good compatibility between yeast and chitosan, and molecular mobility which related to the mechanical property of the films. How can the author prove that? Please explain more, or add more results to support and discuss.

Response 11: Studies on the interactions of chitosan molecules with β-glucans of the yeast cell wall were added to the paper. This information was related to the improvement of the mechanical resistance of the films.

The following information was added: “Similarly, González-Estrada et al. (2015) added D. hansenni yeast to Covalently Cross-Linked Arabinoxylans Films. The evaluation of the mechanical properties of the films showed that tensile strength, elongation at break and Young's modulus values decreased when D. hansenii was added in the film. . They reported that changes in the mechanical properties were due to defects in the arabinoxylans films which contributes to an early rupture of the films during tensile tests.

β-glucans are the main structural components of the cell wall in yeasts. D. hansinni cell wall was characterized by Medina-Córdova et al. (2018) using NMR. They found structures containing (1-6)-branched (1-3)-β-D-glucan. The interaction between chitosan and β-glucans in films have been previously reported. Koc et al. (2020) studied the interactions between a fungal extract and polyphenols in chitosan matrix. FTIR spectra indicated that the OH absorption peak was broadened and shifted towards lower frequencies due to the formation of hydrogen bonds between the chitosan film matrix, β-glucan and polyphenolic compounds. In our study, stress at break increased with the addition of yeast, which could indicate the presence of interactions and good compatibility between molecules (e.g. chitosan molecules and yeast β-glucans). Thus, enhancing mechanical resistance of the films.

References:

González-Estrada, R.R.; Calderón-Santoyo, M.; Carvajal-Millan, E.; Ascencio Valle, F.J.; Ragazzo-Sánchez, J.A.; Brown-Bojorquez F.; Rascón-Chu, A. Covalently Cross-Linked Arabinoxylans Films for Debaryomyces hansenii Entrapment. Molecules 2015, 20, 11373-11386.

Medina-Córdova, N.; Reyes-Becerril, M.; Ascencio, F.; Castellanos, T.; Campa-Córdova, A.I.; Angulo, C. Immunostimulant effects and potential application of β-glucans derived from marine yeast Debaryomyces hansenii in goat peripheral blood leucocytes. Int J Biol Macromol., 116: 599-606.

Point 12: In conclusion, which one is the suggested optimal formulation of film for future application?

Response 12: Considering the results we added the following paragraph: “The combined use of chitosan at 0.5 and 1.0% with the antagonistic yeast need to be tested in vivo trials on citrus fruit, in order to determine if the resulting population dynamics is sufficient to prevent fungal infection by Penicillium italicum.”

Reviewer 2 Report

The present study evaluated the fungal inhibition (mycelial growth, germination, and sporulation) of Penicillium italicum and the effect in the physicochemical properties of chitosan films incorporated with Debaryomyces hansenii. I found the study concise, coherent with the goal, and with a certain degree of novelty. The manuscript is professionally written, and the conclusions are well-supported by the results obtained. However, some aspects need attention before the publication process can proceed. The comments and suggestions are in the attached version. 

Author Response

Response to Reviewer 2 Comments

Point 1: Explain!.

Response 1: According to the viability results the population dynamics was maintained during the nine days of evaluation.

Point 2: What about the risk of diffusion into the food matrix?

Response 2: This point was not analyzed, however as a comment the film-forming solutions were already tested on Persian limes as coatings with good results for preserving fruit quality and protection against the blue mold agent P. italicum.

Point 3: Alphabetical order

Response 3: the keywords were reordered as suggested by the reviewer.

Point 4: Please check the citation style!.

Response 4: the citation style was changed according to the journal.

Point 5: Are there studies with chitosan?

Response 5: To our knowledge there´s no report about the addition of antagonistic microorganisms into chitosan matrices for film formation, but there are a few reports with the incorporation of microbial antagonists on chitosan-based coatings. That´s why we decided to add information about the use of other types of biopolymers for microbial entrappment.

Point 6: What is special about this yeast?

Response 6: The following information was added: “. Marine yeast Debaryomyces hansenii can control phytopathogens throughout several mechanisms of action such as synthesis of hydrolytic enzymes (chitinase, glucanase, protease), competition for space and nutrients, production of killer toxins and biofilms onto the fruit surface, thus the use of antagonistic D. hansenii to manage fungal diseases is a promising alternative to chemical management [17–19].”

Point 7: Molecular identification?

Response 7: The pathogen used in this investigation belongs to the microbial collection of CIBNOR, in their laboratory years ago made the molecular identifaction of the citrus pathogen. The following information was added: “The antagonistic yeast and the pathogen Penicillium italicum (isolated from Persian lime) were provided by CIBNOR S.C. The yeast was maintained on yeast extract peptone dextrose (YPD) medium [23]. The fungus was maintained on PDA (Potato dextrose agar) plates (Sigma-Aldrich, St. Louis, MO, USA).” For a better comprehension about the origin of the pathogen.

Point 8: How did you know that the yeast was still alive? The acidic medium might kill the yeast. Please, explain.

Response 8: We followed the protocol proposed by Homez-Jara et al. (2018), one step involves the partial elimination of the acetic acid by vacuum filtration, this action allows the formation of an edible film without presence of acetic acid, in a preliminar test when the film-forming solution was processed without vacuum filtration the resulting films were opaque and the migration of acetic acid was evidenced. On the other hand, with the population dynamics assessments only viable cells can growth on artificial medium, besides the entrapped cells exhibited biological activity against the fungus.

Point 9: Please, include the equation for the reader's comprehension.

Response 9: The equation was added: Inhibition (%) = [(dc —dt)/dc] x 100

dc (cm) is the mean of colony diameter for the control and dt (cm) is the mean of the colony diameter of treatments

Point 10: What are the units of the Y exe for each figure?

Response 10: The missing information was added as suggested by the reviewer.

Point 11: I believe that this paragraph needs to be removed from here and added to line 189

Response 11: The paragraph was changed as suggested by the reviewer.

Point 12: It is not possible that yeasts also eat the chitosan? Did you try any test to see this likely scenario?

Response 12: We didn´t any test taking into account that scenario, we observed that films are in a good conditions during storage (about one year), the films exhibited with flexibility, no presence of cracks or other sign of deterioration.

Point 13: Which formulation, according to your results, is recommendable?

Response 13: Considering the results we added the following paragraph: “The combined use of chitosan at 0.5 and 1.0% with the antagonistic yeast need to be tested in vivo trials on citrus fruit, in order to determine if the resulting population dynamics is sufficient to prevent fungal infection by Penicillium italicum.”.

Reviewer 3 Report

Abstract: Please amend the repetitions by replacing the words „elaborated” and „evaluated”

Introduction: More information concerning the active packaging based on chitosan has to be provided.

Film formation: Please explain how the film-forming solution was sterilised? Moreover, clearly indicate why Tween 80 was added.

It is not clear if the amounts of glycerol and Tween 80 were calculated in relation to chitosan or solvent. This requires further explanation.

Section 2.3. Film preparation: There is no information concerning the quantity of the yeast pellets used in the procedure. Moreover, the size of Petri dishes has to be indicated.

Lines 109-110: The Authors indicate that “The opacity was calculated based on average thickness.” Please provide more details.

Please explain why the opacity was analysed at 600 nm.

The Authors indicate that antifungal activity depends on the concentration of chitosan. In my opinion, it depends on the chitosan/yeast ratio, and it has to be calculated and presented in the work.

An explanation is necessary why in the case of the sample containing a higher amount of chitosan the addition of yeast improved antifungal properties and how the entrapping of yeast was established. How does the antifungal mechanism of the obtained materials work?

The claim that yeast significantly influences mechanical properties is largely unfounded. The changes in the structure seem to be essential in this case. For this reason, an FTIR analysis has to be performed and described.

The colour change parameter should be calculated and discussed.

Discussion: “Chitosan inhibition could be related to the electrostatic forces between the amino groups (-NH2) of chitosan with the negative charges on the cell membrane of the spore, causing changes in its permeability and alterations at the intracellular level, affecting the germination proces” (lines 235-237) Taking into account the quoted statement please explain why the inhibition was not observed with the increase in the chitosan concentration.

Lines 265-266 “The increase in stress at break is related to good interaction between polymers, im- proving mechanical resistance of the films” What type of polymers do the Authors have in mind?

Lines 270-271 “This behavior was attributed to the interaction between the chitosan polymer chains with the fungal extract molecules.” It has to be explained what kind of interaction can be observed.

Please explain why the strain at break increased in the case of samples with an addition of yeast named 0.5 and 1% while a decrease was observed in the case of the 1,5% sample?

Conclusions:  The Authors claim that formed films “becoming a potential treatment for fungal diseases in citrus fruits.” The conclusion is unfounded and has to be justified.

Author Response

Response to Reviewer 3 Comments

Point 1: Abstract: Please amend the repetitions by replacing the words „elaborated” and „evaluated”

Response 1: the changes were made as suggested by the reviewer.

Point 2: Introduction: More information concerning the active packaging based on chitosan has to be provided.

Response 2: The following information was added: “Several studies reported the carrier capacity of chitosan of several compounds such as gallic acid (Zarandona et al., 2020), nisin (Zimet et al., 2019), sulfur nanoparticles (Shankar & Rhim, 2018), essential oils (Lian et al., 2019), among others. However, to our knowledge, there´s no report about the incorporation of marine yeasts Debaryomyces hansenii in chitosan-based films.”

  1. Zarandona, I.; Puertas, A.I.; Dueñas, M.T.; Guerrero, P.; de la Caba, K. Assessment of Active Chitosan Films Incorporated with Gallic Acid. Food Hydrocolloids 2020, 101, 105486.
  2. Zimet, P.; Mombrú, Á.W.; Mombrú, D.; Castro, A.; Villanueva, J.P.; Pardo, H.; Rufo, C. Physico-Chemical and Antilisterial Properties of Nisin-Incorporated Chitosan/Carboxymethyl Chitosan Films. Carbohydrate polymers 2019, 219, 334–343.
  3. Shankar, S.; Rhim, J.-W. Preparation of Sulfur Nanoparticle-Incorporated Antimicrobial Chitosan Films. Food Hydrocolloids 2018, 82, 116–123.
  4. Lian, H.; Peng, Y.; Shi, J.; Wang, Q. Effect of Emulsifier Hydrophilic-Lipophilic Balance (HLB) on the Release of Thyme Essential Oil from Chitosan Films. Food Hydrocolloids 2019, 97, 105213.

Point 3: Film formation: Please explain how the film-forming solution was sterilised? Moreover, clearly indicate why Tween 80 was added.

Response 3: The following paragraph was added: “The film forming solutions (without yeast) were sterilized at 121 °C during 15 min.” Tween 80 was added to the formulation thinking in the future application of the film-forming solutions on citrus, Tween 80 as surfactant reduce the superficial tension of the film-forming solutions favouring a better distribution and adherence of the biopolymer on treated fruit.

Point 4: It is not clear if the amounts of glycerol and Tween 80 were calculated in relation to chitosan or solvent. This requires further explanation.

Response 4: The quantity of Gly and Tween 80 was added in relation to solvent.

Point 5: Film preparation: There is no information concerning the quantity of the yeast pellets used in the procedure. Moreover, the size of Petri dishes has to be indicated.

Response 5: More details about the procedure were added to the paragraph “. For the films incorporated with the antagonistic yeast, firstly a solution containing the yeast was prepared: briefly, in a biosafety hood (Novatech BBS-DDC) Petri dishes con-taining yeasts were scrapped with a sterile inoculation loop, then the yeast were incor-porated to a sterile saline solution (0.85%) and adjusted to 1107 cell/mL. Then, the solution was centrifuged at 6000 rpm during 20 min, the resulting pellet was aseptically removed and incorporated to the sterile chitosan-based film forming solution in sterile conditions. Finally, the film forming solutions with and without yeast were poured into Petri dishes (60x15 mm) and they were left to dry 40 h in an oven at 40ºC (22% of relative humidity)” Depending the volume of the chitosan-based film forming solution the solution containing the yeasts was prepared. The information of the Petri dishes was added.

Point 6: Lines 109-110: The Authors indicate that “The opacity was calculated based on average thickness.” Please provide more details.

Response 6: The paragraph was moved to the “Opacity” assessment to provide more details of the test.

Point 7: Please explain why the opacity was analysed at 600 nm.

Response 7: The human eye is receptive to electromagnetic radiation that we call visible light, the range of values that goes from 350nm (nanometers) to 780nm visible light. The distribution of the eye's sensitivity to different wavelengths is bell-shaped with a maximum value for cones around 600 nm and a maximum for rods around 500 nm. Thus, the methodology for light-visible (VIS) barrier properties of the films is estandarized to 600 nm, for human eye perception.

Point 8: The Authors indicate that antifungal activity depends on the concentration of chitosan. In my opinion, it depends on the chitosan/yeast ratio, and it has to be calculated and presented in the work.

Response 8: According to our results and the statistical design we observed an enhancing efficacy as chitosan concentration increased in presence of yeast, the concentration of antagonistic microorganism do not changed during the experiment that´s why we consider this effect.

Point 9: An explanation is necessary why in the case of the sample containing a higher amount of chitosan the addition of yeast improved antifungal properties and how the entrapping of yeast was established. How does the antifungal mechanism of the obtained materials work?

Response 9: The following information was added: “The results showed a better efficacy by the combination of chitosan with the antagonistic yeast (Fig 1), previous studies have reported a synergistic effect of chitosan against pathogens when the biopolymer is combined with antagonistic yeasts, due to a better distribution of the antagonistic cells that can be in contact with the pathogen allowing the production of lytic enzymes, as previously reported (Aloui et al., 2014; González-Estrada et al., 2017; Wang et al., 2019; Yu et al., 2007; Zhou et al., 2016).”

References:

González-Estrada, R.R.; Carvajal-Millán, E.; Ragazzo-Sánchez, J.A.; Bautista-Rosales, P.U.; Calderón-Santoyo, M. Control of Blue Mold Decay on Persian Lime: Application of Covalently Cross-Linked Arabinoxylans Bioactive Coatings with Antagonistic Yeast Entrapped. {LWT} - Food Science and Technology 2017, 85, 187–196, doi:10.1016/j.lwt.2017.07.019.

Aloui, H.; Khwaldia, K.; Sánchez‐González, L.; Muneret, L.; Jeandel, C.; Hamdi, M.; Desobry, S. Alginate Coatings Containing Grapefruit Essential Oil or Grapefruit Seed Extract for Grapes Preservation. International Journal of Food Science & Technology 2014, 49, 952–959.

The following information was added: “β-glucans are the main structural components of the cell wall in yeasts. D hansenni cell wall was characterized by Medina-Córdova et al. (Medina-Córdova et al., 2018) using NMR. They found structures containing (1-6)-branched (1-3)-β-D-glucan. The interaction between chitosan and β-glucans in films have been previously reported. Koc et al. (Koc et al., 2020) studied the interactions between a fungal extract and polyphenols in chitosan matrix. FTIR spectra indicated that the OH absorption peak was broadened and shifted towards lower frequencies due to the formation of hydrogen bonds between the chitosan film matrix, β-glucan and polyphenolic compounds. In our study, stress at break increased with the addition of yeast, which could indicate the presence of interactions and good compatibility between molecules (e.g. chitosan molecules and yeast β-glucans). Thus, enhancing mechanical resistance of the films.”

Point 10: The claim that yeast significantly influences mechanical properties is largely unfounded. The changes in the structure seem to be essential in this case. For this reason, an FTIR analysis has to be performed and described.

Response 10: González-Estrada et al. (2015) incorporated D. hansenni into arabinoxylan films and studied the structural changes by means of FTIR. They report that the interactions between the components as well as the adduct formation were not observed by FTIR, which could be explained by the relatively low amount of entrapped cells.

We consider that in order to study the structural changes of the film components, it is necessary to use more specific techniques such as NMR. However, it is not possible for us to do that analysis since we do not have that equipment.

Reference: González-Estrada, R.R.; Calderón-Santoyo, M.; Carvajal-Millan, E.; Ascencio Valle, F.J.; Ragazzo-Sánchez, J.A.; Brown-Bojorquez F.; Rascón-Chu, A. Covalently Cross-Linked Arabinoxylans Films for Debaryomyces hansenii Entrapment. Molecules 2015, 20, 11373-11386.

Point 11: The colour change parameter should be calculated and discussed.

Response 11: The following equation was added:

The following data was also added and disccussed as suggested by the reviewer.

Without yeast

With yeast

17.5±0.47Ca

15.6±1.27Cb

21.4±0.90Ba

22.8±1.11Ba

29.2±0.70Aa

29.12±1.59Aa

Point 12: Discussion: “Chitosan inhibition could be related to the electrostatic forces between the amino groups (-NH2) of chitosan with the negative charges on the cell membrane of the spore, causing changes in its permeability and alterations at the intracellular level, affecting the germination proces” (lines 235-237) Taking into account the quoted statement please explain why the inhibition was not observed with the increase in the chitosan concentration.

Response 12: In a recent study Chávez-Magdaleno et al. (2019) reported a similar behaviour of the strains Colletotrichum acutatum and Colletotrichum gloeosporioides isolated from avocado, besides Cortés-Rivera (2021) reported a similar effect against Rhizopus stolonifer isolated from soursop; these pathogens were not affected by the sole application of chitosan, suggesting that pathogens could be resistant to chitosan effect.

References

Chávez-Magdaleno, M. E., Gutiérrez-Martínez, P., Montaño-Leyva, B., & González-Estrada, R. R. (2019). Evaluación in vitro del quitosano y aceites esenciales para el control de dos especies patógenas de Colletotrichum aisladas de aguacate (Persea americana Mill). TIP. Revista Especializada En Ciencias Químico-Biológicas, 22.

Cortés-Rivera, H. J., González-Estrada, R. R., Huerta-Ocampo, J. Á., Blancas-Benítez, F. J., & Gutiérrez-Martínez, P. (2021). Evaluación de quitosano comercial y extractos acuosos de mesocarpio de coco (Cocos nucifera L.) para el control de Rhizopus stolonifer aislado de guanábana (Annona muricata L.): Pruebas in vitro. TIP Revista Especializada En Ciencias Químico-Biológicas, 24.

Point 13: Lines 265-266 “The increase in stress at break is related to good interaction between polymers, im- proving mechanical resistance of the films” What type of polymers do the Authors have in mind?

Response 13: Studies on the interactions of chitosan molecules with β-glucans of the yeast cell wall were added to the paper. This information was related to the improvement of the mechanical resistance of the films.

The following information was added: “Similarly, González-Estrada et al. (2015) added D. hansenni yeast to Covalently Cross-Linked Arabinoxylans Films. The evaluation of the mechanical properties of the films showed that tensile strength, elongation at break and Young's modulus values decreased when D. hansenii was added in the film. . They reported that changes in the mechanical properties were due to defects in the arabinoxylans films which contributes to an early rupture of the films during tensile tests.

β-glucans are the main structural components of the cell wall in yeasts. D. hansinni cell wall was characterized by Medina-Córdova et al. (2018) using NMR. They found structures containing (1-6)-branched (1-3)-β-D-glucan. The interaction between chitosan and β-glucans in films have been previously reported. Koc et al. (2020) studied the interactions between a fungal extract and polyphenols in chitosan matrix. FTIR spectra indicated that the OH absorption peak was broadened and shifted towards lower frequencies due to the formation of hydrogen bonds between the chitosan film matrix, β-glucan and polyphenolic compounds. In our study, stress at break increased with the addition of yeast, which could indicate the presence of interactions and good compatibility between molecules (e.g. chitosan molecules and yeast β-glucans). Thus, enhancing mechanical resistance of the films.

References:

González-Estrada, R.R.; Calderón-Santoyo, M.; Carvajal-Millan, E.; Ascencio Valle, F.J.; Ragazzo-Sánchez, J.A.; Brown-Bojorquez F.; Rascón-Chu, A. Covalently Cross-Linked Arabinoxylans Films for Debaryomyces hansenii Entrapment. Molecules 2015, 20, 11373-11386.

Medina-Córdova, N.; Reyes-Becerril, M.; Ascencio, F.; Castellanos, T.; Campa-Córdova, A.I.; Angulo, C. Immunostimulant effects and potential application of β-glucans derived from marine yeast Debaryomyces hansenii in goat peripheral blood leucocytes. Int J Biol Macromol., 116: 599-606.

Point 14: Lines 270-271 “This behavior was attributed to the interaction between the chitosan polymer chains with the fungal extract molecules.” It has to be explained what kind of interaction can be observed.

Response 14: Studies on the interactions between chitosan molecules and  β-glucans of the yeast cell wall were added to the paper.

β-glucans are the main structural components of the cell wall in yeasts. D. hansinni cell wall was characterized by Medina-Córdova et al. (2018) using NMR. They found structures containing (1-6)-branched (1-3)-β-D-glucan. The interaction between chitosan and β-glucans in films have been previously reported. Koc et al. (2020) studied the interactions between a fungal extract and polyphenols in chitosan matrix. FTIR spectra indicated that the OH absorption peak was broadened and shifted towards lower frequencies due to the formation of hydrogen bonds between the chitosan film matrix, β-glucan and polyphenolic compounds. In our study, stress at break increased with the addition of yeast, which could indicate the presence of interactions and good compatibility between molecules (e.g. chitosan molecules and yeast β-glucans). Thus, enhancing mechanical resistance of the films.

References: Medina-Córdova, N.; Reyes-Becerril, M.; Ascencio, F.; Castellanos, T.; Campa-Córdova, A.I.; Angulo, C. Immunostimulant effects and potential application of β-glucans derived from marine yeast Debaryomyces hansenii in goat peripheral blood leucocytes. Int J Biol Macromol., 116: 599-606.

Point 15: Please explain why the strain at break increased in the case of samples with an addition of yeast named 0.5 and 1% while a decrease was observed in the case of the 1,5% sample?

Response 15: The following information was added as sugested by reviewer: “ The decrease in strain at break showed by films with 1.5% chitosan + yeast might be due to the higher number of interactions between the yeast and the functional groups in chitosan when the latter polymer is in higher concentration.”

Point 16: Conclusions:  The Authors claim that formed films “becoming a potential treatment for fungal diseases in citrus fruits.” The conclusion is unfounded and has to be justified.

Response 16: The conclusions were changed as follows: “In conclusion, chitosan combined with D. hansenii proved to be more successful in inhibiting the growth of fungi in vitro than the application of chitosan alone. At 0.5 and 1.0% of chitosan concentration the viability of yeast was maintained during 9 days. Incorporation of antagonistic yeast improved the mechanical resistance of the films. Chitosan combination with yeast not only achieves the viability of the antagonist, but also shows important effects on parameters involved in the development and proliferation of the fungus. In addition, the films have desirable characteristics for packaging use on citrus in the future. Nevertheless, the combined use of chitosan at 0.5 and 1.0% with the antagonistic yeast need to be tested in vivo trials on citrus fruit, in order to determine if the resulting population dynamics is sufficient to prevent fungal infection by Penicillium italicum”.

References added:

Aloui, H., Khwaldia, K., Sánchez‐González, L., Muneret, L., Jeandel, C., Hamdi, M., & Desobry, S. (2014). Alginate coatings containing grapefruit essential oil or grapefruit seed extract for grapes preservation. International Journal of Food Science & Technology, 49(4), 952–959.

Chávez-Magdaleno, M. E., Gutiérrez-Martínez, P., Montaño-Leyva, B., & González-Estrada, R. R. (2019). Evaluación in vitro del quitosano y aceites esenciales para el control de dos especies patógenas de Colletotrichum aisladas de aguacate (Persea americana Mill). TIP. Revista Especializada En Ciencias Químico-Biológicas, 22.

Cortés-Rivera, H. J., González-Estrada, R. R., Huerta-Ocampo, J. Á., Blancas-Benítez, F. J., & Gutiérrez-Martínez, P. (2021). Evaluación de quitosano comercial y extractos acuosos de mesocarpio de coco (Cocos nucifera L.) para el control de Rhizopus stolonifer aislado de guanábana (Annona muricata L.): Pruebas in vitro. TIP Revista Especializada En Ciencias Químico-Biológicas, 24.

González-Estrada, R. R., Carvajal-Millán, E., Ragazzo-Sánchez, J. A., Bautista-Rosales, P. U., & Calderón-Santoyo, M. (2017). Control of blue mold decay on Persian lime: Application of covalently cross-linked arabinoxylans bioactive coatings with antagonistic yeast entrapped. {LWT} - Food Science and Technology, 85, 187–196. https://doi.org/10.1016/j.lwt.2017.07.019

Koc, B., Akyuz, L., Cakmak, Y. S., Sargin, I., Salaberria, A. M., Labidi, J., Ilk, S., Cekic, F. O., Akata, I., & Kaya, M. (2020). Production and characterization of chitosan-fungal extract films. Food Bioscience, 35, 100545.

Lian, H., Peng, Y., Shi, J., & Wang, Q. (2019). Effect of emulsifier hydrophilic-lipophilic balance (HLB) on the release of thyme essential oil from chitosan films. Food Hydrocolloids, 97, 105213.

Medina-Córdova, N., Reyes-Becerril, M., Ascencio, F., Castellanos, T., Campa-Córdova, A. I., & Angulo, C. (2018). Immunostimulant effects and potential application of β-glucans derived from marine yeast Debaryomyces hansenii in goat peripheral blood leucocytes. International Journal of Biological Macromolecules, 116, 599–606.

Shankar, S., & Rhim, J.-W. (2018). Preparation of sulfur nanoparticle-incorporated antimicrobial chitosan films. Food Hydrocolloids, 82, 116–123.

Wang, F., Deng, J., Jiao, J., Lu, Y., Yang, L., & Shi, Z. (2019). The combined effects of Carboxymethyl chitosan and Cryptococcus laurentii treatment on postharvest blue mold caused by Penicillium italicum in grapefruit fruit. Scientia Horticulturae, 253, 35–41. https://doi.org/https://doi.org/10.1016/j.scienta.2019.04.031

Yu, T., Li, H. Y., & Zheng, X. D. (2007). Synergistic effect of chitosan and Cryptococcus laurentii on inhibition of Penicillium expansum infections. International Journal of Food Microbiology, 114(3), 261–266.

Zarandona, I., Puertas, A. I., Dueñas, M. T., Guerrero, P., & de la Caba, K. (2020). Assessment of active chitosan films incorporated with gallic acid. Food Hydrocolloids, 101, 105486.

Zhou, Y., Zhang, L., & Zeng, K. (2016). Efficacy of Pichia membranaefaciens combined with chitosan against Colletotrichum gloeosporioides in citrus fruits and possible modes of action. Biological Control, 96, 39–47.

Zimet, P., Mombrú, Á. W., Mombrú, D., Castro, A., Villanueva, J. P., Pardo, H., & Rufo, C. (2019). Physico-chemical and antilisterial properties of nisin-incorporated chitosan/carboxymethyl chitosan films. Carbohydrate Polymers, 219, 334–343.

Round 2

Reviewer 3 Report

I have to admit that the Authors have significantly improved the submitted manuscript.